# ENFORCING PHYSICS-BASED ALGEBRAIC CONSTRAINTS FOR INFERENCE OF PDE MODELS ON UNSTRUCTURED GRIDS

## ABSTRACT

The lack of relevant physical constraints in data-driven models of physical systems, such as neural network parameterized partial differential equations (PDEs), might lead to unrealistic modeling outcomes. A majority of approaches to solving this problem are based on forcing a model to satisfy a set of equations representing physical constraints. Currently available approaches can enforce a very limited set of constraints and are applicable only to uniform spatial grids. We propose a method for enforcing general pointwise, differential and integral constraints on unstructured spatial grids. Our method is based on representing a model's output in terms of a function approximation and enforcing constraints on that approximation. We demonstrate wide applicability and strong performance of our approach in data-driven learning of dynamical PDE systems and distributions of physical fields.

## 1 INTRODUCTION

Multiple works have shown the capability of neural networks to solve complex physical problems and learn the behavior of physical systems from data. Examples include learning and solving ordinary differential equations (ODEs) [6], partial differential equations (PDEs) [28; 20] and rigid body dynamics [31; 5]. Purely data-driven models are typically not forced to satisfy physical constraints of the system that generated the data. This might lead to unrealistic predictions that violate some known properties of the underlying physical system.

Incorporation of relevant constraints allows to make a better use of the available data and makes predictions more physically plausible. The field dealing with physics-constrained learning is diverse and offers many approaches to adding constraints to models. We refer the reader to many reviews for details [30; 3; 36; 19]. The approach we consider in this work is based on forcing a model to satisfy algebraic constraints represented by a set of equalities and inequalities. This is the most commonly used approach which allows to represent a wide range of constraints and has been shown to work well in many cases [18; 17; 25]. However, while many constraints can be represented algebraically, it is not always clear how to evaluate and enforce them.

Currently available approaches to enforcing algebraic constraints are limited to uniform grids and have a very narrow range of constraints they can enforce (e.g. only pointwise, or specific differential constraints), see Section 5 for details of related work. Such approaches can be readily applied to models based on convolutional neural networks (CNNs) but cannot be extended to recently developed models based on graph neural networks (GNNs) [33; 27; 15] and other models working on unstructured grids.

We propose a much more general method which allows to enforce pointwise, differential and integral constraints on unstructured spatial grids and demonstrate its applicability in learning of PDE-driven dynamical systems and distributions of physical fields. The method is based on using a models's output at the nodes of a grid to construct an interpolant and applying constraints directly to that interpolant (Section 3).

Code and data will be made publicly available.

## 2    BACKGROUND

**PDE-driven dynamical systems.**    Many physical systems can be described in terms of PDEs. Such systems are defined on a bounded domain on which they evolve over time. We consider continuous dynamical systems with state $u(t, \boldsymbol{x}) \in \mathbb{R}^p$ that evolves over time $t \in \mathbb{R}_{\geq 0}$ and spatial locations $\boldsymbol{x} \in \Omega \subset \mathbb{R}^D$. For physical systems, $D$ is typically limited to $\{1, 2, 3\}$ although our method will work with any value of $D$. We assume the system is governed by an unknown PDE

$$\frac{\partial u(t, \boldsymbol{x})}{\partial t} = F(\boldsymbol{x}, u(t, \boldsymbol{x}), \nabla_{\boldsymbol{x}} u(t, \boldsymbol{x}), \nabla_{\boldsymbol{x}}^2 u(t, \boldsymbol{x}), ...) \tag{1}$$

which describes the temporal evolution of the system in terms of the locations $\boldsymbol{x}$, state $u$ and its first and higher-order partial derivatives w.r.t. $\boldsymbol{x}$. The goal of a data-driven PDE model is to learn the dynamics $F$ from data.

Data for learning $F$ is collected by measuring the state of the system at observation locations $(\boldsymbol{x}_1, \ldots, \boldsymbol{x}_N)$ over increasing time points $(t_0, \ldots, t_M)$. This results in a dataset $(\mathbf{y}(t_0), \ldots, \mathbf{y}(t_M))$, where $\mathbf{y}(t_i) = (u(t_i, \boldsymbol{x}_1), \ldots, u(t_i, \boldsymbol{x}_N))$ is a collection of observations. The dataset is used to train the model to predict $(\mathbf{y}(t_1), \ldots, \mathbf{y}(t_M))$ starting from the initial state $\mathbf{y}(t_0)$. Training is typically done by minimizing an average loss between the model's predictions $\mathbf{u}(t)$ and the data $\mathbf{y}(t)$.

PDE models differ in restrictions they impose on time points (temporal grid) and observation locations (spatial grid). Some models require both grids to be uniform [23], other models relax these requirements and allow arbitrary spatial [27] and spatio-temporal grids [15]. We build our algebraic constraints method using the model from [15] as the most general one. The model is based on application of the method of lines [32] to Equation 1 which results into a system of ODEs

$$\dot{\mathbf{u}}(t) := \begin{pmatrix} \frac{du(t, \boldsymbol{x}_1)}{dt} \\ \vdots \\ \frac{du(t, \boldsymbol{x}_N)}{dt} \end{pmatrix} \approx \begin{pmatrix} F_\theta(\boldsymbol{x}_1, \boldsymbol{x}_{\mathcal{N}(1)}, u_1, u_{\mathcal{N}(1)}) \\ \vdots \\ F_\theta(\boldsymbol{x}_N, \boldsymbol{x}_{\mathcal{N}(N)}, u_N, u_{\mathcal{N}(N)}) \end{pmatrix} \tag{2}$$

which approximates the solution of Equation 1 at the observation locations $\boldsymbol{x}_i$ using their neighboring points $\mathcal{N}(i)$, where $\boldsymbol{x}_{\mathcal{N}(i)}$ and $u_{\mathcal{N}(i)}$ are the neighbors' positions and states respectively, and $u_i$ is $u(t, \boldsymbol{x}_i)$. The approximate solution converges to the true solution as $N$ increases. The true dynamics $F$ is approximated by a parametric model $F_\theta$ whose parameters $\theta$ are learned by minimizing the difference between the model's predictions

$$\mathbf{u}(t) = \mathbf{u}(0) + \int_0^t \dot{\mathbf{u}}(\tau) d\tau \tag{3}$$

and the data $\mathbf{y}(t)$. The integral in Equation 3 is solved using a numerical ODE solver. In [15], the function $F_\theta$ was represented by a graph neural network (GNN) which takes states and locations at an observation point $i$ and its neighboring points $\mathcal{N}(i)$. The observation points are connected into a grid using Delaunay triangulation which allows to naturally define $\mathcal{N}(i)$ as a set of points connected to the point $i$. However, $F_\theta$ can be represented by other models and a different neighbor selection criterion can be used. The model parameters $\theta$ are learned by minimizing the MSE between $\mathbf{y}(t)$ and $\mathbf{u}(t)$

$$L_{\text{data}} = \frac{1}{M} \sum_{i=1}^{M} \|\mathbf{u}(t_i) - \mathbf{y}(t_i)\|_2^2. \tag{4}$$

The gradient of $L_{\text{data}}$ w.r.t. $\theta$ is evaluated using the adjoint method as shown in [7].

**Generative Adversarial Networks**    One of the tasks that we consider is learning distributions of physical fields. For that purpose we utilize generative adversarial networks (GANs). A GAN is a generative model consisting of a generator and a discriminator [12]. The generator, $G$, learns to transform a random variable $Z \sim p_Z$ over a latent space $\mathcal{Z}$ to the data space $\mathcal{Y}$ in such a way that the discriminator, $D$, cannot tell the difference between samples generated by $G$ and samples from the data distribution $p_{\text{data}}$. Both, $G$ and $D$ are learned by solving the following minimax problem

$$\min_G \max_D V(G, D) = \mathbb{E}_{Y \sim p_{\text{data}}} [\log D(Y)] + \mathbb{E}_{Z \sim p_Z} [\log (1 - D(G(Z)))]. \tag{5}$$

Solution of this problem exists and is unique with the optimal generator perfectly mimicking the data distribution [12].

## 3 METHODS

In this section we presents an approach to evaluating pointwise, differential and integral constraints on unstructured grids. Then, we demonstrate how this approach can be used to enforce arbitrary soft and linear hard constraints.

### 3.1 EVALUATING CONSTRAINTS ON UNSTRUCTURED GRIDS

We assume the data $\mathbf{y}(t)$ is available at observation points $(\boldsymbol{x}_1, \dots, \boldsymbol{x}_N)$ and time points $(t_1, \dots, t_M)$ and that a model makes predictions $\mathbf{u}(t)$ at these points. We assume the predictions to be evaluations of an unknown underlying function. Since the underlying function is unknown, we cannot impose constraints on it directly. Instead, we approximate it by an interpolant $u_f(t, \boldsymbol{x})$ and impose constraints on $u_f(t, \boldsymbol{x})$ (Figure 1). The approximation is constructed from $\mathbf{u}(t)$ by placing a basis function at each $\boldsymbol{x}_i$ and representing $u_f(t, \boldsymbol{x})$ as

$$u_f(t, \boldsymbol{x}) = \sum_{j=1}^{N} \alpha_j(t)\phi_j(\boldsymbol{x}), \tag{6}$$

where $\phi_j$ is a scalar basis function at $\boldsymbol{x}_j$ and $\alpha_j \in \mathbb{R}^p$. The coefficients $\alpha_j(t)$ are obtained from $\mathbf{u}(t)$ (see Section 3.4).

Next, we show how to evaluate constraints on $u_f(t, \boldsymbol{x})$ using basic building blocks. To avoid cluttered notation, we consider equality constraints and assume $u(t, \boldsymbol{x}), \boldsymbol{x} \in \mathbb{R}$. Generalization to inequality constraints, vector fields and higher spatial dimensions is straightforward.

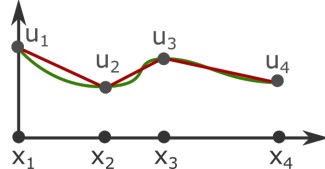

Figure 1: Example of approximating an unknown underlying function (green) by an interpolant (red) constructed from observations $u_1, \dots, u_4$.

**Pointwise constraints.** Consider points $\mathbf{z} = (\mathbf{z}_1, \dots, \mathbf{z}_K)$ in $\Omega$ on which a pointwise constraint $h(u_f(t, \mathbf{z}_i)) = 0$ should be evaluated. Assume the function $h : \mathbb{R} \to \mathbb{R}$ is representable in terms of a finite number of functions $\gamma_m(u_f(t, \mathbf{z}_i)) : \mathbb{R} \to \mathbb{R}$ indexed by $m$. For example, should the constraint be $h(u_f) = 3u_f + u_f^2 = 0$, then we would define $\gamma_1(u_f) = u_f$, $\gamma_2(u_f) = u_f^2$ and $h(u_f) = 3 \cdot \gamma_1(u_f) + \gamma_2(u_f) = 0$. Then, $h$ can be evaluated by evaluating each $\gamma_m$ as

$$\gamma_m(u_f(t, \mathbf{z}_i)) = \gamma_m\left(\sum_{j=1}^{N} \alpha_j(t)\phi_j(\mathbf{z}_i)\right) = \gamma_m\left(\Phi_{i,\cdot}\boldsymbol{\alpha}(t)\right), \tag{7}$$

where $\boldsymbol{\alpha}(t) = (\alpha_1(t), \dots, \alpha_N(t))^T$, $\Phi$ is $K$-by-$N$ matrix with elements $\Phi_{i,j} = \phi_j(\mathbf{z}_i)$, and $\Phi_{i,\cdot}$ is the $i$'th row of $\Phi$.

**Differential constraints.** Consider the same setup as before but now $h(u_f(t, \mathbf{z}_i)) = 0$ consists of differential operators and is representable in terms of a finite number of functions $\frac{\partial^q \gamma_m(u_f(t, \mathbf{z}_i))}{\partial \mathbf{z}_i^q}$ : $\mathbb{R} \to \mathbb{R}$ indexed by $m$, where the derivative order $q$ could be different for each $m$. For example, should the constraint be $h(u_f) = 3u_f + u_f \cdot \frac{\partial u_f^2}{\partial \boldsymbol{x}} = 0$, then we would define $\gamma_1(u_f) = u_f$, $\gamma_2(u_f) = u_f^2$ and $h(u_f) = 3 \cdot \gamma_1(u_f) + \gamma_1(u_f) \cdot \frac{\partial \gamma_2(u_f)}{\partial \mathbf{z}} = 0$. Then, $h$ can be evaluated by evaluating each $\frac{\partial^q \gamma_m(u_f(t, \mathbf{z}_i))}{\partial \mathbf{z}_i^q}$ using the generalization of the chain rule (Appendix A) which contains only two types of terms. The first type of terms $\frac{d\gamma_m}{du_f}, \dots, \frac{d^q \gamma_m}{du_f^q}$ can be evaluated using automatic differentiation while the second type of terms $\frac{\partial u_f}{\partial \mathbf{z}_i}, \dots, \frac{\partial^q u_f}{\partial \mathbf{z}_i^q}$ can be evaluated as

$$\frac{\partial^q u_f}{\partial \mathbf{z}_i^q} = \sum_{j=1}^{N} \alpha_j(t)\frac{\partial^q \phi_j(\mathbf{z}_i)}{\partial \mathbf{z}_i^q} = \Phi_{i,\cdot}^{(q)}\boldsymbol{\alpha}(t), \tag{8}$$

where $\Phi_{i,j}^{(q)} = \frac{\partial^q \phi_j(\mathbf{z}_i)}{\partial \mathbf{z}_i^q}$. Mixed partial derivatives can be handled in a similar way (Appendix A).

**Integral constraints.** Consider the same setup as before but with $h(u_f(t, \boldsymbol{x})) = \int_\Omega \tau(u_f(t, \boldsymbol{x}))d\boldsymbol{x} = 0$, where the function $\tau : \mathbb{R} \to \mathbb{R}$ is representable in terms of functions $\gamma_m(u_f(t, \mathbf{z}_i)) : \mathbb{R} \to \mathbb{R}$ similarly to the pointwise constraints. Then, $\int_\Omega \tau(u_f(t, \boldsymbol{x}))d\boldsymbol{x}$ can be evaluated using a numerical integration technique, e.g. midpoint rule, Gaussian quadrature or Monte-Carlo integration, as

$$\int_\Omega \tau(u_f(t, \boldsymbol{x}))d\boldsymbol{x} \approx \sum_{i=1}^K \tau(u_f(t, \mathbf{z}_i))\mu_i, \tag{9}$$

where $K$ is the number of integration points, $\mu_i$ are integration coefficients which depend on the grid and integration method, and $\tau(u_f(t, \mathbf{z}_i))$ is evaluated as in Equation 7.

## 3.2 SOFT CONSTRAINTS

Soft constraints are implemented by minimizing the following loss $L_{\text{data}} + \lambda r(h(u_f))$, where $\lambda \in \mathbb{R}$ and $L_{\text{data}}$ is defined as in Equation 4. We set $r(h(u_f)) = \frac{1}{KM}\sum_{i=1}^K\sum_{j=1}^M h(u_f(t_j, \mathbf{z}_i))^2$ for pointwise and differential constraints and $r(h(u_f)) = \frac{1}{M}\sum_{j=1}^M h(u_f(t_j, \boldsymbol{x}))^2$ for integral constraints.

## 3.3 HARD CONSTRAINTS

Our method allows to implement hard constraints by projecting the interpolant $u_f(t, \boldsymbol{x})$ to a subset of functions which satisfy the required constraints. Namely, if $u_f(t, \boldsymbol{x})$ does not satisfy constraints $g$ and $h$, it is projected to a subset of functions which satisfy the constraint by solving the following optimization problem

$$\begin{aligned} \min_{\hat{u}_f \in V_\phi} \quad & \|u_f - \hat{u}_f\|_{L^2}^2 \\ \text{s.t.} \quad & h(\hat{u}_f) = 0, \\ & g(\hat{u}_f) \leq 0, \end{aligned} \tag{10}$$

where the projection is denoted by $\hat{u}_f(t, \boldsymbol{x})$ and $V_\phi$ is spanned by the basis functions.

Using the basis representation $u_f(t, \boldsymbol{x}) = \sum_{i=1}^N \alpha_i(t)\phi_i(\boldsymbol{x})$ and $\hat{u}_f(t, \boldsymbol{x}) = \sum_{i=1}^N \beta_i(t)\phi_i(\boldsymbol{x})$ we can rewrite the optimization problem (10) as

$$\begin{aligned} \min_{\boldsymbol{\beta}(t) \in \mathbb{R}^N} \quad & (\boldsymbol{\alpha}(t) - \boldsymbol{\beta}(t))^T \hat{\Phi}(\boldsymbol{\alpha}(t) - \boldsymbol{\beta}(t)) \\ \text{s.t.} \quad & h(\hat{u}_f) = 0, \\ & g(\hat{u}_f) \leq 0, \end{aligned} \tag{11}$$

where $\boldsymbol{\beta}(t) = (\beta_1(t), \ldots, \beta_N(t))^T$ and $\hat{\Phi}_{i,j} = \int_\Omega \phi_i(\boldsymbol{x})\phi_j(\boldsymbol{x})d\boldsymbol{x}$.

To train the model end-to-end, the problem (11) should be differentiable. Agrawal et. al. [1] proposed differentiable convex optimization which could be used in this case if the problem (11) could be expressed in a DPP-compliant way (see [1]). To do that, we restrict ourselves to constraints that can be expressed as an equality or inequality between $A\boldsymbol{\beta}(t)$ and $\boldsymbol{b}$, where $A$ is a constant matrix and $\boldsymbol{b}$ is a constant vector. This formulation admits pointwise, differential and integral constraints on untransformed $u_f$. The objective function is convex since its Hessian is positive-semidefinite i.e. for any $v \in \mathbb{R}^N$

$$v^T\hat{\Phi}v = \sum_{i,j=1}^N v_iv_j\hat{\Phi}_{i,j} = \sum_{i,j=1}^N \langle v_i\phi_i, v_j\phi_j \rangle_{L^2} = \langle \sum_{i=1}^N v_i\phi_i, \sum_{j=1}^N v_j\phi_j \rangle_{L^2} \geq 0. \tag{12}$$

This allows to solve the problem (11) and differentiate its solution $\boldsymbol{\beta}^*(t)$ w.r.t. $\boldsymbol{\alpha}(t)$. The model parameters are found by minimizing the following loss function $L_{\text{data}} + \lambda L_{\text{proj}}$, where $\lambda \in \mathbb{R}$ and $L_{\text{data}}$ is defined as in Equation 4 but with $\mathbf{u}(t_i)$ replaced by $\hat{\mathbf{u}}(t_i) = (\hat{u}_f(t_i, \boldsymbol{x}_1), \ldots, \hat{u}_f(t_i, \boldsymbol{x}_N))$. We set $L_{\text{proj}} = \frac{1}{NM}\sum_{i=1}^N\sum_{j=1}^M \|u_f(t_j, \boldsymbol{x}_i) - \hat{u}_f(t_j, \boldsymbol{x}_i)\|_2^2$. The second term makes the optimization procedure prefer models that predict $u_f$ close to the feasible set of the problem (11).

We note that the proposed approach is currently limited to small-scale problems due to existing computational bottlenecks in the implementation of differentiable convex optimization [1].

### 3.4 BASIS FUNCTIONS

Selecting appropriate basis is crucial for efficiency and applicability of the proposed method. Ideally, the basis should allow efficient construction of $u_f(t, \boldsymbol{x})$ from $\mathbf{u}(t)$, contain no tunable parameters, and lead to sparse matrices $\Phi, \Phi^{(q)}$ and $\hat{\Phi}$. We consider bases from two families: Lagrange basis functions and radial basis functions (RBFs).

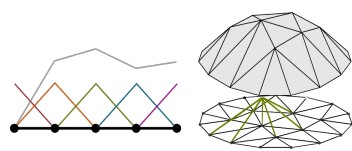
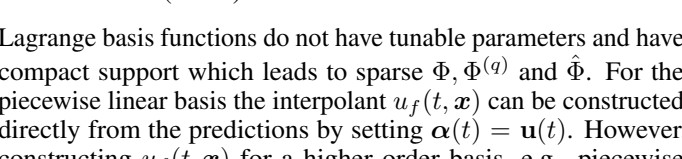

Lagrange basis functions do not have tunable parameters and have compact support which leads to sparse $\Phi, \Phi^{(q)}$ and $\hat{\Phi}$. For the piecewise linear basis the interpolant $u_f(t, \boldsymbol{x})$ can be constructed directly from the predictions by setting $\boldsymbol{\alpha}(t) = \mathbf{u}(t)$. However, constructing $u_f(t, \boldsymbol{x})$ for a higher order basis, e.g. piecewise

Figure 2: 1D and 2D piecewise linear basis functions (colored) and function built from them (grey). Black dots represent observation points.

quadratic, requires the model to make predictions not only at the observation points, but also at some extra points where the data is not available. In Section 4 we demonstrate one approach to solving this problem. After extending the state $\mathbf{u}(t)$ by predictions at the extra nodes, the coefficients $\boldsymbol{\alpha}(t)$ can be evaluated similarly to the piecewise linear basis. In this work we use piecewise linear (PWL) and piecewise quadratic (PWQ) bases. Examples of PWL basis functions are shown in Figure 2.

Radial basis functions have a wider range of properties. Some RBFs have tunable parameters, some don't. The matrices $\Phi, \Phi^{(q)}$ and $\hat{\Phi}$ evaluated with RBFs are typically dense, but RBFs with compact support exist (e.g. bump function). The interpolant $u_f(t, \boldsymbol{x})$ can be constructed by evaluating $\boldsymbol{\alpha}(t) = \boldsymbol{K}^{-1}\mathbf{u}(t)$, where $\boldsymbol{K}^{-1}$ is the inverse of the interpolation matrix of the given RBF and $\boldsymbol{K}_{ij} = \phi(\|\boldsymbol{x}_i - \boldsymbol{x}_j\|)$, where $\phi$ is an RBF and $\boldsymbol{x}_i, \boldsymbol{x}_j$ are observation locations. In this work we use the cubic RBF basis i.e. $\phi(r) = r^3$.

We use PyTorch [26] to handle sparse matrices and to evaluate $\boldsymbol{K}^{-1}\mathbf{u}(t)$ in a differentiable way.

## 4 EXPERIMENTS

In the following experiments we use the relative error between the data $\mathbf{y}(t)$ and model predictions $\mathbf{u}(t)$ defined as $\frac{\|\mathbf{y}(t) - \mathbf{u}(t)\|_2}{\|\mathbf{y}(t)\|_2}$ and consider only soft constraints. We present an experiment with hard constraints implemented as shown in Section 3.3 in Appendix D. Data generation is described in Appendix B. Training, testing and modeling details are in Appendix C. All experiments were run on a single NVIDIA Quadro P5000 GPU. All errors bars represent one standard deviation of the results over five random seeds.

### 4.1 REPLACING EXISTING METHODS

In this experiment we take existing models which incorporate physics-based constraints in training and replace their constraint enforcing approaches with ours. We consider two works. First, [37] which trains a GAN to produce divergence-free vector fields using zero-divergence constraint. Second, [10] which predicts warping fields driving the evolution of sea surface temperature by observing snapshots of the temperature over time while enforcing gradient and divergence constraints on the warping fields (see Appendix C for more details). Both models work on uniform grids which allows them to evaluate constraints using finite differences. For comparison, we replace finite differences with our method and observe how it changes the models' performance. In both cases we use the PWL basis.

For [37] we track the mean divergence and discriminator loss. Results of the original approach are as follows: mean divergence $0.079$ and discriminator loss $0.091$. With our method the mean divergence was $0.014$ and the discriminator loss was $0.088$. Both approaches results in similar discriminator losses but our approach produces a smaller mean divergence (smaller is better). Our method increased the runtime per epoch by 6%.

For [10] we track the total, divergence and smoothness losses which, with the original approach, were $0.139, 8.4 \cdot 10^{-5}$ and $1.51 \cdot 10^{-4}$, respectively. With our approach the losses were $0.139, 8.3 \cdot 10^{-5}$ and $1.51 \cdot 10^{-4}$, respectively. Both methods produce very similar results. Our method increased the runtime per epoch by 30%.

Overall, replacing existing constraint enforcing approaches by ours on data from uniform grids resulted in comparable model performance, except for runtime which was slightly increased.

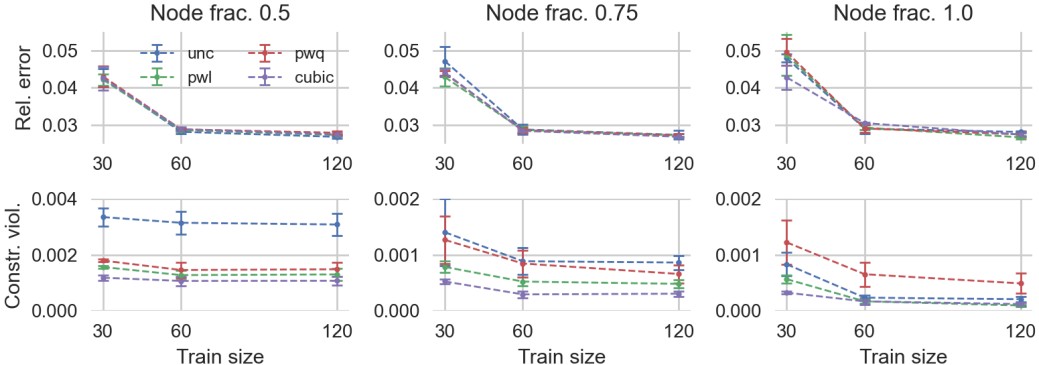

Figure 4: Effects of amount of data and grid sizes on relative errors and constraint violations for the Cahn-Hilliard equation. All results are for the test set. Constraint violations are evaluated as the mean absolute violation of the constraint, $|\int_\Omega u_f(t, \boldsymbol{x})d\boldsymbol{x} - C|$ over all simulations and time points. In most simulations $C \approx 0.5$.

## 4.2 CAHN-HILLIARD EQUATION WITH AN INTEGRAL CONSTRAINT

We start with the 1D Cahn-Hilliard equation

$$\frac{\partial u}{\partial t} = 2\nabla^2(u(1-u)^2 - u^2(1-u) - \epsilon^2\nabla^2 u) \tag{13}$$

which is known to conserve the state $u$ i.e. $h(u) = \int_\Omega u(t, x)dx - C = 0$ at all time points, where $C = \int_\Omega u(0, x)dx$ is a constant. This is an example of a conservation law which are abundant in nature and are important class of constraints that data-driven models of physical systems should satisfy. Conservation laws can be expressed in differential and integral forms and this experiment demonstrates how the integral form can be enforced. The constraint is evaluated using the midpoint rule as shown in the previous section with a single $\gamma_1$ being the identity function. We use PWL, PWQ and cubic RBF bases and compare the results to an unconstrained model.

For training we use 30, 60 and 120 simulations while the test set consist of 60 simulations. Simulations in the training/test data last for 0.0015/0.0030 seconds and contain 50/100 uniformly spaced time points. The full spatial grid consists of 101 uniformly spaced nodes. We randomly sample 50%, 75% and 100% of the nodes and train/test on the resulting (irregular) spatial grid. Training and testing is done with identical spatial grids. An example of a spatial grid with 50% of nodes is shown in Figure 3. We evaluate the constraint on a uniform grid with 200 nodes placed on top of the original grid.

Figure 3: 1D spatial grid for the Cahn-Hilliard equation.

To learn the dynamics of the system we use the model from [15] (Section 2). We found that using a GNN produced poor results. For that reason we represented the function $F_\theta$ with a multiplayer perceptron (MLP) which updates the state of each node based on the states of all other nodes in the grid (results for a GNN are in Appendix E). The MLP contains two hidden layers with Leaky ReLU nonlinearities. The number of hidden neurons was set to the number of nodes in the grid.

The coefficients $\boldsymbol{\alpha}(t)$ for the PWL and cubic bases can be evaluated directly from the model predictions at the grid nodes. But the PWQ basis requires extra predictions to be available between the nodes. This is problematic since there is no data at these points to guide the model's predictions. To solve this problem we introduce a small MLP which is applied to consecutive pairs of nodes. The MLP takes the states at both nodes and the distance between them as the input and estimates the state at the midpoint between the two nodes. The MLP is trained jointly with the main model and uses only the constraint-related loss term during training.

For testing, we construct the interpolant $u_f(t, \boldsymbol{x})$ using the thin plate spline basis ($\phi(r) = r^2\log r$) and evaluate the constraint on that interpolant. This allows to make a fair comparison between the unconstrained model and different bases and avoid biasing or ovefitting to bases used for training.

Figure 4 shows results of the experiment. We observe that changing the node fraction does not significantly affect the relative errors but has noticeable effect on constraint violations, especially for the unconstrained model. Constrained models tend to show similar or better performance than the unconstrained model. Among all bases, the cubic basis consistently results in lower relative errors and constraint violations. However, the simpler PWL basis often performs on par with the cubic basis, especially on denser spatial grids. We also observe that coarsening of the grid increases the constraint violation gap between constrained and unconstrained models and that this gap seems to not close as we increase the amount of training data.

The PWQ basis performs rather poorly on fine grids which is likely due to a suboptimal approach to evaluating the state at the extra nodes. A better approach could consider not only pairs of points but also larger neighborhoods. Nonetheless, the PWQ basis achieves good performance on coarse grids which shows that piecewise bases of higher order could potentially be used to enforce constraints. This will allow to scale to grids with a large number of nodes due to sparsity of the constraint matrices and efficient evaluation of $\boldsymbol{\alpha}$.

### 4.3 HEAT EQUATION WITH A MONOTONICITY CONSTRAINT

We impose constraints on a 2D system governed by the heat equation $\frac{\partial u}{\partial t} = \nabla^2 u$ for which the generated initial conditions (ICs) are monotone in one direction. Since the ICs are monotone, the state $u$ remains monotone at all time points as well. We enforce the monotonicity constraint as $\frac{\partial u}{\partial x} \geq 0$. The constraint is evaluated as shown in the previous section with $\gamma_1$ being the identity function.

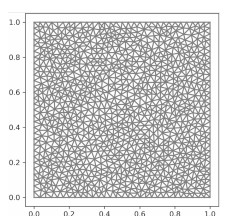

For training we use 15, 30 and 90 simulations while the test set consist of 120 simulations. Simulations in the training/test data last for 0.1/0.2 seconds and contain 21/41 uniformly spaced time points. The full spatial grid consists of 1087 nodes. We randomly sample 33%, 66% and 100% of the nodes and train/test on the resulting (irregular) spatial grid. Training and testing

Figure 5: Grid for the heat equation.

is done with identical spatial grids. Spatial grid with 100% of nodes is shown in Figure 5. The constraint is evaluated at the nodes of a uniform $51 \times 51$ grid placed on top of the original grid.

To learn the dynamics of the system we use the model from [15] directly with the messaging and aggregation networks being MLPs with a single hidden layer consisting of 60 neurons with Tanh nonlinearities and the input/output sizes of 4/40 and 41/1 respectively.

During testing, we use predictions of the models to construct an interpolant $u_f(t, \boldsymbol{x})$ using the thin plate spline basis and evaluate the constraint on that interpolant. This allows to make a fair comparison between the unconstrained model and different bases.

Figure 7 shows results of the experiment. We observe that changing the node fraction equally increases relative errors of all models and has noticeable effect on constraint violations, especially for the unconstrained model. Constrained models tend to show slightly higher or comparable relative errors but noticeably lower constraint violations than the unconstrained model. The cubic and PWL bases perform equally well in this case. Similarly to the experiment in the previous section, we observe that coarsening of the

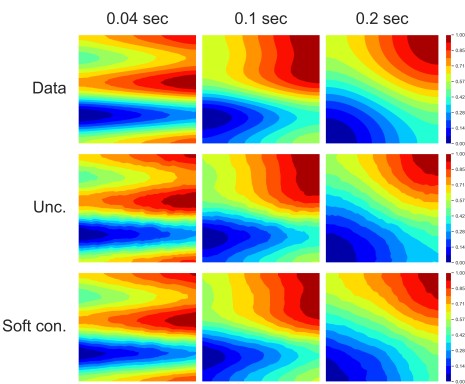

Figure 6: Comparison of data with predictions of unconstrained and constrained models trained on 30 simulations, full spatial grid and using PWL basis for the constrained model. The predictions are for a test case.

grid introduces a larger constraint violation gap between constrained and unconstrained models and that this gap seems to not close as we increase the amount of training data.

Figure 6 shows qualitative difference between predictions of constrained and unconstrained models. It can be noted that predictions from the constrained model have noticeably smoother contours thus making the field more monotonous in the horizontal direction.

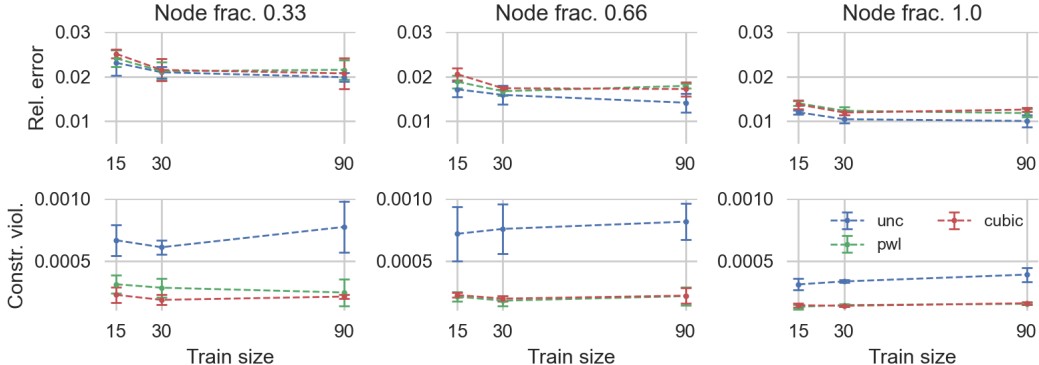

Figure 7: Effects of amount of data and grid sizes on relative errors and constraint violations for the heat equation. Results are for the test set. Constraint violations are evaluated as the mean absolute violation of the constraint $\frac{\partial u_f}{\partial x} \geq 0$ over all simulations and time points.

## 4.4 LEARNING DISTRIBUTIONS OF PHYSICAL FIELDS

We demonstrate the effect of adding constraints to a GAN when learning distributions of physical fields on unstructured grids. We use Wasserstein GAN (WGAN) [2] as a more stable variant of a GAN. We use MLPs as a generator and discriminator. Unconstrained and constrained models are trained for 1.2M iterations. Constraints are enabled only after 600k iterations. Constrained models are trained similarly to the unconstrained ones but with a modified generator loss defined as $L_G + \lambda \ln (1 + L_C)$, where $L_G$ is the standard generator loss and $L_C$ is the constraint-based loss. We define $L_C$ as the mean value of $h(u)^2$, where $h$ is a constraint evaluated at the centroid of each cell in the grid.

### 4.4.1 ZERO-DIVERGENCE FIELDS

Divergence-free vector fields are often encountered in solutions of fluid dynamics problems. The divergence-free constraint on a vector field $u(x, y) = (u_1(x, y), u_2(x, y))^T$ is defined as $h(u) = \frac{\partial u_1}{\partial x} + \frac{\partial u_2}{\partial y} = 0$. The constraint is enforced using the PWL basis. We generated a dataset with 10k divergence-free fields on an unstructured grid with 1050 nodes (Figure 14)

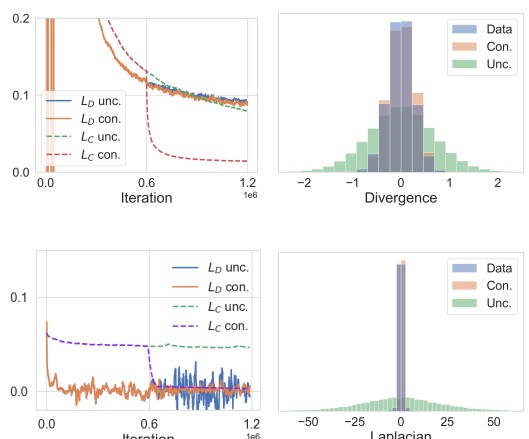

Figure 8: First row: $L_D$, $L_C$ and histograms of divergences of samples from data, constrained and unconstrained WGANs. Second row: $L_D$, $L_C$ and histograms of Laplacians of samples from data, constrained and unconstrained WGANs. Constraints are evaluated at cell centroids.

and used a WGAN to learn a distribution over such fields. Note that the generated fields are not entirely divergence-free but have small residual divergence due to discretization errors.

Figure 9a shows that there is a clear difference in the quality of the samples generated by the unconstrained and constrained models. Samples from the constrained model are smoother and more similar to the data. Quantitative comparison of the samples presented in Figure 8 shows that the constrained model generates fields that have much lower constraint violation and divergence distribution very similar to that of the data.

### 4.4.2 ZERO-LAPLACIAN FIELDS

Fields with zero Laplacian represent solutions to some PDEs, for example the steady-state heat equation. The zero-Laplacian constraint on a scalar field $u(x, y)$ is defined as $h(u) = \frac{\partial^2 u}{\partial x^2} + \frac{\partial^2 u}{\partial y^2} = 0$. The constraint is enforced using the cubic basis as the PWL basis has zero second derivatives

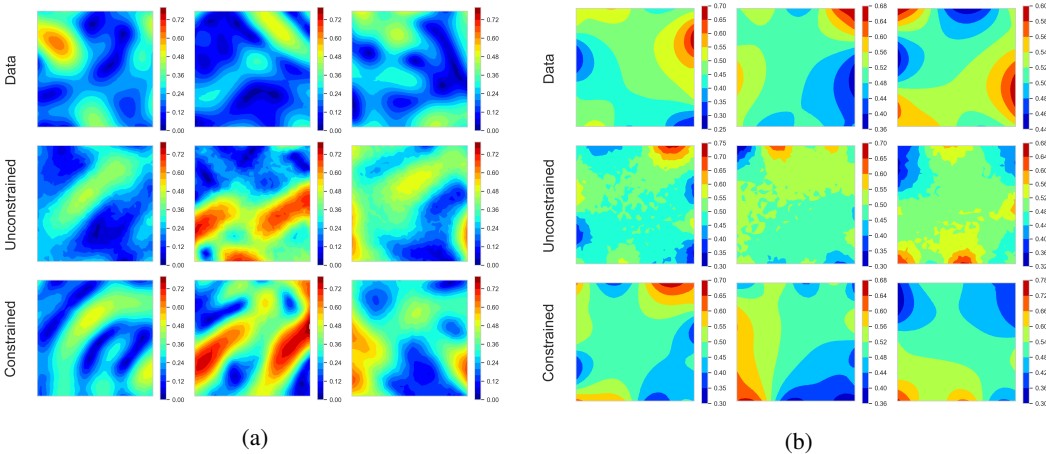

Figure 9: Magnitudes of random samples from the dataset, unconstrained and constrained WGANs. a) zero-divergence fields, b) zero-Laplacian fields.

everywhere. We generated a dataset with 10k Laplacian-free fields on an unstructured grid with 1050 nodes (Figure 14) and used a WGAN to learn a distribution over such fields. Note that the generated fields are not entirely Laplacian-free due to discretization errors.

Results of the experiment are shown in Figures 9b and 8. Similarly to the divergence-free case, visual quality of the fields generated by the constrained model is significantly better than for the unconstrained model. Quantitative comparison of the samples presented in Figure 8 shows that the constrained model generates fields that have much lower constraint violation and Laplacian distribution very similar to that of the data.

## 5  RELATED WORK

**Soft constraints.**   Soft constraints are widely used due to being relatively easy to implement. Examples include lake temperature prediction [18; 16], traffic simulation [22], fluid and climate modeling [11; 10; 4], where constraints are evaluated pointwise or using finite differences.

**Hard constraints.**   Approaches to implementing hard constraints are diverse and can be categorized as processing the output of an unconstrained model [4; 17; 24; 34] and designing a model that produces feasible predictions by default [23; 25; 14; 9; 13; 8; 38].

**Constrained PDE models**   Current approaches to enforcing soft [10; 11] and hard [21; 25; 17] constraints are limited to specific types of constraints and spatial grids. For example, [25; 17] implement only hard differential constraints and both are limited to uniform grids. Uniform grids allow to evaluate constraints efficiently e.g. using finite differences [10; 21; 25] or fast Fourier transform [17] but assuming that the data lies on a uniform grid might be limiting.

**Constrained GANs**   Works such as [37; 17] showed how physics-based constraints benefit training and quality of the generated samples but are also limited to uniform grids.

## 6  CONCLUSION

We presented a general approach to enforcing algebraic constraints on unstructured grids and showed how it can be used to enforce soft and hard constraints. We demonstrated applicability of the approach to learning of PDE-driven dynamical systems and distributions of physical fields. We considered two families of basis functions for constructing the interpolant and showed how Lagrange basis functions of order higher than one can be used. Our method allows to drop the unrealistic assumption about uniformity of spatial grids and shows promising results on various tasks.

## REPRODUCIBILITY STATEMENT

All details required to reproduce the experiments are provided in Section 4 and Appendices. Code and data used to run the experiments will be made publicly available after the review process.

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

## A GENERALIZED CHAIN RULE AND HANDLING MIXED PARTIAL DERIVATIVES.

Let $y = g(x_1, \ldots, x_n)$ with all arguments being either identical, distinct or grouped. Then, partial derivatives of $f(y)$ can be evaluated using the Faà di Bruno's formula

$$\frac{\partial^n f(y)}{\partial x_1 \cdots \partial x_n} = \sum_{\pi \in \Pi} f^{(|\pi|)}(y) \prod_{B \in \pi} \frac{\partial^{|B|} y}{\prod_{j \in B} \partial x_j},$$

where $\Pi$ is the set of all partitions of the set $\{1, \ldots, n\}$, $B$ runs through elements of the partition $\pi$, $f^{(m)}$ denotes $m$'th derivative, and $|\cdot|$ is cardinality.

The formula consists of two terms: $f^{(|\pi|)}(y)$, which can be evaluated using automatic differentiation, and $\frac{\partial^{|B|} y}{\prod_{j \in B} \partial x_j}$, which can be evaluated as shown in Equation 8.

In case that all $x_1, \ldots, x_n$ are identical, the mixed derivative $\frac{\partial^n f(y)}{\partial x_1 \cdots \partial x_n}$ reduces to $\frac{\partial^n f(y)}{\partial x_1^n}$.

## B DATA GENERATION

In all cases we run simulation on a fine grid and then interpolate the results to a coarser grid represented as the "full grid" in the experiments.

### B.1 CAHN-HILLIARD EQUATION WITH AN INTEGRAL CONSTRAINT

Training and testing data was obtained by solving

$$\frac{\partial u}{\partial t} = 2\nabla^2(u(1-u)^2 - u^2(1-u) - \epsilon^2 \nabla^2 u) \tag{14}$$

on a unit interval with periodic boundary conditions and $\epsilon = 0.04$. The domain was represented by a uniform grid with 100 nodes and the time step was set to 1.0e-6 sec. The initial conditions $u_0(x)$ were generated as follows

$$\tilde{u}_0(x) = \sum_{i=1}^{10} (\lambda_i \cos((x-s)2\pi) + \gamma_i \sin((x-s)2\pi)) + \frac{\lambda_0}{2}, \tag{15}$$

$$u_0(x) = \frac{\tilde{u}_0(x) - \min \tilde{u}_0(x)}{\max \tilde{u}_0(x) - \min \tilde{u}_0(x)}, \tag{16}$$

where $\lambda_i, \gamma_i \sim \text{Unif}(-1, 1)$ and $s \sim \text{Unif}(0, 1)$.

Examples of the simulations are shown in Figure 10.

### B.2 HEAT EQUATION WITH A MONOTONICITY CONSTRAINT

Training and testing data was obtained by solving

$$\frac{\partial u}{\partial t} = D\nabla^2 u \tag{17}$$

on a unit square with zero Neumann boundary conditions and $D = 0.2$. The domain was represented by an unstructured grid with 2971 nodes and the time step was set to 0.001 sec. The initial conditions $u_0(x)$ were generated as

$$f(x) = \sum_{i=0}^{6} \omega_i x^i, \tag{18}$$

$$g(y) = \frac{1}{2} \sum_{i=1}^{3} (\lambda_i \cos((x+s)2\pi) + \gamma_i \sin((x+s)2\pi)) + \frac{\lambda_0}{2}, \tag{19}$$

$$\tilde{u}_0(x, y) = f(x) + g(y), \tag{20}$$

$$u_0(x, y) = \frac{\tilde{u}_0(x, y) - \min \tilde{u}_0(x, y)}{\max \tilde{u}_0(x, y) - \min \tilde{u}_0(x, y)}, \tag{21}$$

where $\omega_i \sim \text{Unif}(0.1, 1.1)$, $\lambda_i, \gamma_i, s \sim \text{Unif}(-1, 1)$.

Examples of the simulations are shown in Figure 11.

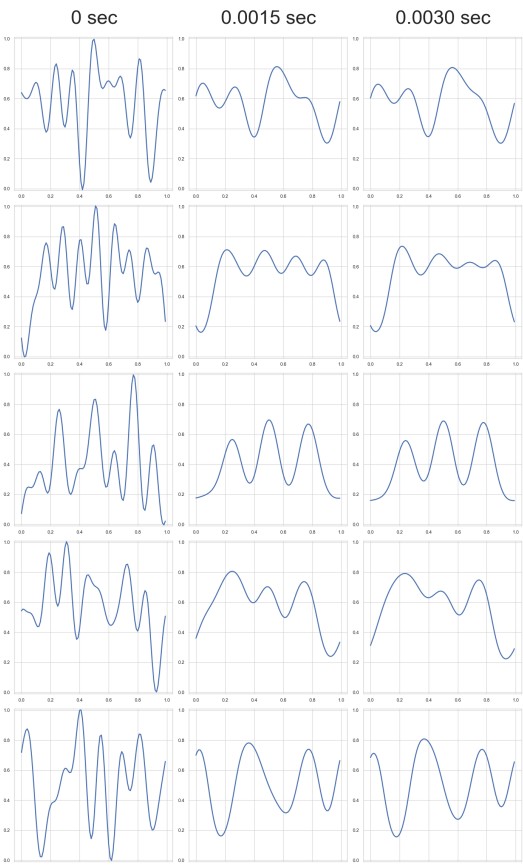

Figure 10: Examples of simulations for the Cahn-Hilliard equation on a unit interval.

## B.3  GAN WITH A DIVERGENCE CONSTRAINT

The data was generated by sampling random velocity fields and then projecting them to the space of divergence-free fields. The procedure was as follows. First, a random velocity field $u_0(x, y)$ was generated on a unit square by generating each component $i$ as

$$\tilde{u}_{0i}(x, y) = \sum_{k,l=-N}^{N} \lambda_{kl} \cos(kx + ly) + \gamma_{kl} \sin(kx + ly), \tag{22}$$

$$u_{0i}(x, y) = 6 \times \left( \frac{\tilde{u}_{0i}(x, y) - \min \tilde{u}_{0i}(x, y)}{\max \tilde{u}_{0i}(x, y) - \min \tilde{u}_{0i}(x, y)} - 0.5 \right), \tag{23}$$

where $N = 10$ and $\lambda_{kl}, \gamma_{kl} \sim \mathcal{N}(0, 1)$. Then, the divergence-free component of $u_0(x, y)$, denoted by $u_0^*(x, y)$, was extracted by using the projection method by solving $\nabla \cdot u_0 = \nabla^2 \phi$ for $\phi$ and then evaluating $u_0^*(x, y) = u_0(x, y) - \nabla \phi$. Finally, the data was scaled to $[-1, 1]$.

## B.4  GAN WITH A LAPLACIAN CONSTRAINT

The data was generated by solving

$$\nabla^2 u = 0 \tag{24}$$

on a unit square with Dirichlet boundary conditions. The domain was represented by an unstructured grid with 2971 nodes. The boundary conditions were generated by generating random functions $u_0(x)$ and using their boundary values as the boundary conditions. The functions $u_0(x)$ were generated as

$$u_0(x, y) = \sum_{k,l=-N}^{N} \lambda_{kl} \cos(kx + ly) + \gamma_{kl} \sin(kx + ly) \tag{25}$$

where $N = 5$ and $\lambda_{kl}, \gamma_{kl} \sim \mathcal{N}(0, 1)$. The data was then scaled to $[0, 1]$.

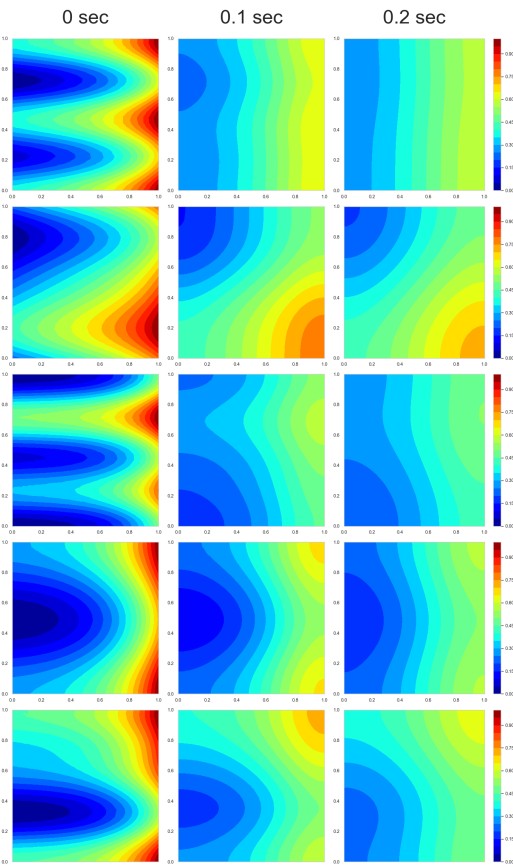

Figure 11: Examples of simulations for the heat equation on a unit square.

## C    MODELS, TRAINING AND TESTING

### C.1    REPLACING EXISTING METHODS

For our comparisons we considered experiments from two works. Next, we provide some details about these experiments.

The first experiment was taken from [37] Section 3.2. The experiment shows how soft physics-based constraints affect predictions of a GAN learning a distribution of divergence-free fields. The data is generated on a uniform grid which allows to evaluate divergence using finite differences. The constraint is enforced through an extra loss term which penalizes violation of the constraint. The performance metric used is the Frobenius norm of the divergence averaged over all fields in a batch. For training we used code provided by the authors with original parameters. We replaced finite differences in the constrained evaluation function with our method.

The second experiment was taken from [10]. This work deals with the task of predicting sea surface temperatures at future times given snapshots of the temperature over current and previous times. The model proposed by the authors accomplishes this tasks by taking a sequence of surface temperatures at times $t_{i-k}, \ldots, t_i$ and predicting the underlying motion field which is then used to predict the temperature at time $t_{i+1}$. Insights about physical properties of the motion field were used to constrain the model's predictions. Constraints are imposed on divergence, magnitude and gradients of the motion field. The data is generated on a uniform grid which allows to evaluate the constraints using finite differences. The constraints are enforced through extra loss terms which penalize violation of the constraints. Performance metrics that were used are MSE between the data and model predictions, smoothness loss and divergence loss. For training we used code provided by the authors with original parameters. We replaced finite differences in the constrained evaluation function with our method.

### C.2 CAHN-HILLIARD EQUATION WITH AN INTEGRAL CONSTRAINT

In all experiments with the Cahn-Hilliard equation we represent the dynamics function $F_\theta$ by an MLP with 2 hidden layers and LeakyReLU nonlinearities (negative slope 0.2). The number of neurons in each layer was set to the number of nodes in the spatial grid on which the model was trained. The predictions $\mathbf{u}(t)$ were obtained by simulating the system forward in time using adaptive Heun solver from torchdiffeq package [6] with rtol and atol set to 1.0e-5 and 1.0e-5 respectively. All models were trained for 1500 epochs using Rprop optimizer [29] with learning rate set to $1.0 \cdot 10^{-6}$ and batch size set to the number of simulations in the training set. Mean squared error was used as the loss function. Spatial and temporal grids in the testing data were the same as in the training data. We set $\lambda = 2$.

### C.3 HEAT EQUATION WITH A MONOTONICITY CONSTRAINT

In all experiments with the heat equation we represent the dynamics function $F_\theta$ by a GNN with the messaging and aggregation networks being MLPs with a single hidden layer consisting of 60 neurons with Tanh nonlinearities and the input/output sizes of 4/40 and 41/1 respectively. The predictions $\mathbf{u}(t)$ were obtained by simulating the system forward in time using adaptive Heun solver from torchdiffeq package [6] with rtol and atol set to 1.0e-5 and 1.0e-5 respectively. All models were trained for 750 epochs using Rprop optimizer [29] with learning rate set to $1.0 \cdot 10^{-6}$ and batch size set to the number of simulations in the training set. Mean squared error was used as the loss function. Spatial and temporal grids in the testing data were the same as in the training data. We set $\lambda = 0.1$.

### C.4 LEARNING DISTRIBUTIONS OF PHYSICAL FIELDS

In both cases we used identical architectures and training process for the constrained and unconstrained models. Both models were trained for 1.2M iterations using the same random seed. Constraints in the constrained model were enabled only after 600k iterations. The base distribution was set to a 128-dimensional isotropic standard normal. Models were trained using RMSProp optimizer [35] with batch size and learning rate set to 64 and 0.00001 respectively. The discriminator's weights were clipped to $[-0.01, 0.01]$.

#### C.4.1 ZERO-DIVERGENCE FIELDS

We used MLPs as a discriminator and generator. The discriminator consisted of 3 hidden layers of sizes 1024-512-256 with LeakyReLU nonlinearities (negative slope 0.2) and input/output size of 2010 and 1 respectively. The generator consisted of 3 hidden layers of sizes 256-512-1024 with LeakyReLU nonlinearities (negative slope 0.2), input/output size of 128 and 2010 respectively, and a final hyperbolic tangent nonlinearity applied to the output. We set $\lambda = 0.2$.

#### C.4.2 ZERO-LAPLACIAN FIELDS

We used MLPs as a discriminator and generator. The discriminator consisted of 3 hidden layers of sizes 1024-512-256 with LeakyReLU nonlinearities (negative slope 0.2) and input/output size of 1086 and 1 respectively. The generator consisted of 3 hidden layers of sizes 256-512-1024 with LeakyReLU nonlinearities (negative slope 0.2), input/output size of 128 and 1086 respectively, and sigmoid function applied to the output. We set $\lambda = 0.0075$.

## D CAHN-HILLIARD EQUATION WITH HARD INTEGRAL CONSTRAINTS

Here we demonstrate how the approach to enforcing hard constraints described in Section 3.3 can be used to enforce integral constraints on a nonuniform grid.

We use the same setup as in Section 4.2 with 30 training simulations and 50% of nodes in the grid. We compare three models: unconstrained model, model with soft constraint and model with hard constraint. We use the PWL basis during training and testing.

Table 1 shows that relative errors of all three models are practically similar but, as Figure 12 demonstrates, constraint violations differ significantly. We see that constraint violations of the model with hard constraint are zero at all time points as expected.

Being able to produce predictions that satisfy some constraints exactly might be very useful for some applications, however, as we mention in Section 3.3, currently this approach to enforcing hard constraints is limited to systems with a relatively small number of nodes and is significantly slower than models with soft constraints. We report training times in Table 1.

Table 1: Test relative errors and training times for the Cahn-Hilliard equation.

| Model | Test rel. err | Training time |
|---|---|---|
| Unc. | $0.041 \pm 0.002$ | 15 min. |
| Soft-con. | $0.044 \pm 0.002$ | 17 min. |
| Hard-con. | $0.043 \pm 0.002$ | 207 min. |

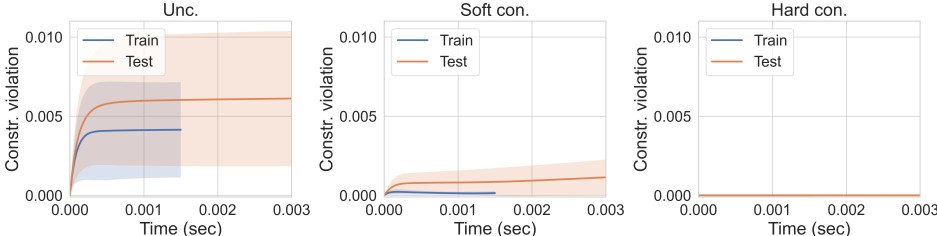

Figure 12: Constraint violation plots for the Cahn-Hilliard equation. Each panel shows the mean absolute violation of the constraint, $|\int_\Omega u_f(t, \boldsymbol{x})d\boldsymbol{x} - C|$, over all train/test simulations for each time point.

## E  LEARNING CAHN-HILLIARD EQUATION WITH GNNS

We use the same setup as in Section 4.2 with 75% of nodes in the grid but trained the models for 1500 epochs. Instead of an MLP we use a GNN with messaging and aggregation networks being MLPs with two hidden layers of size 64 and LeakyReLU nonlinearities (negative slope 0.2). For each node, the GNN evaluates the output as

$$\frac{du_i}{dt} = \gamma \left( \frac{1}{|\mathcal{N}(i)|} \sum_{j \in \mathcal{N}(i)}^{N} \phi(u_i^{\text{proj}}, u_j^{\text{proj}}, x_{ij}^{\text{proj}}), u_i^{\text{proj}} \right), \tag{26}$$

where $u_i^{\text{proj}}$ and $u_j^{\text{proj}}$ are linear projections of the state at nodes $i$ and $j$ and $x_{ij}^{\text{proj}}$ is a linear projection of the pair consisting of the distance between nodes $i$ and $j$ and a unit vector pointing from node $i$ to node $j$. All projections have dimension 16.

We compare constrained and unconstrained models. We use the PWL, PWQ and cubic RBF bases. Results of the experiment are shown in Figure 13. The figure shows that relative errors and constraint violations of all models are significantly higher than for MLP-based models.

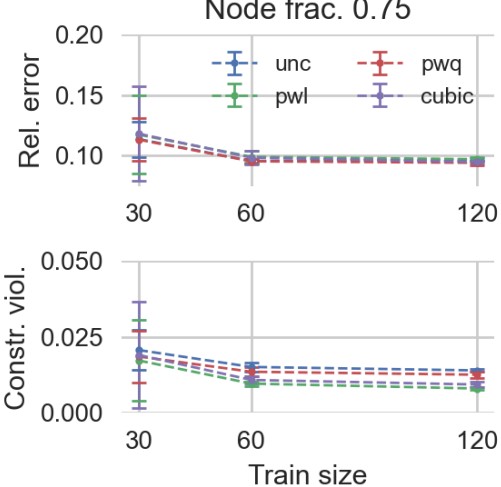

Figure 13: Relative errors and constraint violations for a GNN trained on the Cahn-Hilliard equation. All resuts are for the test set.

## F   EXTRA FIGURES

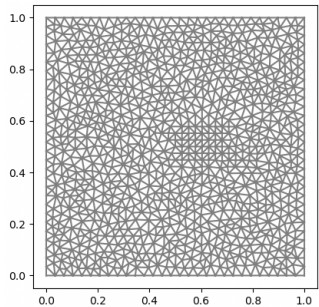

Figure 14: Grid used for training GANs.

