# OpenReview forum: "Enforcing physics-based algebraic constraints for inference of PDE models on unstructured grids"
_ICLR.cc/2022/Conference — ICLR 2022 Submitted_

### Official Review · Reviewer_HR2b · 2021-10-27

**Correctness:** 3
**Technical Novelty And Significance:** 2
**Empirical Novelty And Significance:** 2
**Recommendation:** 5
**Confidence:** 4

**Main Review:**

The paper presents a series of tests which follow a somewhat unintuitive order. First, a regular grid case is presented, which seems to be taken from previous work. Not much detail is given, and the test only confirms that the constraints give very similar results. This test seems like a pure debugging case, as it is based on Cartesian grids instead of the unstructured grids which the paper wants to focus on.

The subsequent sections (4.1 ff) then provide examples with unstructured grids. While the first CH-case is very simplistic, the latter cases contain interesting setups and show some interesting results. E.g., I found it interesting to see how the physical constraints affect the smoothness of the zero divergence fields

In terms of writing, I found the order and argumentation of the results sub-optimal, as mentioned above. The structured grid case seems mostly out of place. The introduction also sounds strange to me, with current work being "limited" to uniform grids. Previous works have admittedly focused on these, but the methods are still applicable - after all, this submission is applying the methods from uniform grids in a fairly straight forward manner. I would recommend to rephrase this. Also, stylistically, citations shouldn't be used as nouns, and the related work seems to brief to me (e.g., generative models for physical problems are missing).


**Summary Of The Paper:**

This paper proposes a method to enforce physical constraints in deep learning models. It provides a nice summary of local, differential and integral constraints, and frames them in a Lagrangian setting. For a reason which I could not clearly follow, the paper focuses on GAN early on. This is intuitive to me, as the physical constraints could nicely stand on their own.


**Summary Of The Review:**

Overall, I think this submission would benefit from further revisions. It is definintely an interesting direction, but in its current state the paper applies existing methods from regular grids to Lagrangian discretizations. This is a good idea, but not a very fundamental step forward. In combination, the results do not contain many surprises, and the focus on GANs in a way detracts from the main goals of the paper. The presentation of the work could also be further improved. So overall, I'm leaning towards the negative side given the current state of the submission.

---

> ### Author Response · Authors · 2021-11-17
> **Response to Reviewer HR2b.**
>
> Thank you for your review. Below, we address the raised questions and concerns.
>
> **Q1:** For a reason which I could not clearly follow, the paper focuses on GAN early on.
> **A1:** GANs can be used for generation of physical fields (as we show in Section 4.4), which is an interesting use case for applying constraints.
>
> **Q2:** Why compare to other methods on regular grids (Section 4.1)?
> **A2:** We believe that it is important to demonstrate that despite being more general and flexible, our method still performs on par with the current methods that can be applied only to regular grids.
>
> **Q3:** The introduction also sounds strange to me, with current work being "limited" to uniform grids. Previous works have admittedly focused on these, but the methods are still applicable - after all, this submission is applying the methods from uniform grids in a fairly straight forward manner.
> **A3:** We kindly ask the reviewer to clarify this statement. To the best of our knowledge, none of the previous works can be extended to unstructured grids. We respectfully disagree: the statement that we are "applying the methods from uniform grids" is not correct. In contrary to previous methods, we deal with interpolants directly, and any constraint evaluation method on a uniform grid can be derived from an interpolant but not vice versa thus making our approach strictly more general.
>
> **Q4:** The related work seems to brief to me (e.g., generative models for physical problems are missing).
> **A4:** We tried to include all relevant works including examples of generative models for physical problems (see Sec.5, "Constrained GANs"). We would be glad to extend the section in case we missed something.

---

> > ### Comment · Reviewer_HR2b · 2021-11-23
> > **Re: Response**
> >
> > Thanks for replying in detail to the questions and issues raised in the reviews.
> >
> > However, in light of the uniform assessment from the other reviewers, I still think that my original assessment is appropriate: the paper is not fully ready yet for publication at ICLR. I hope the authors can continue to improve their submission to make a more convincing case for their approach.

---

### Official Review · Reviewer_3bQB · 2021-10-29

**Correctness:** 3
**Technical Novelty And Significance:** 3
**Empirical Novelty And Significance:** 3
**Recommendation:** 5
**Confidence:** 4

**Main Review:**

The task addressed by the authors is crucial in machine learning since purely data driven approaches blatantly fail at conserving several physical quantities such as momentum or energy. In particular, the enforcement of algebraic constraints (whether differential or integral) on a learned model is a necessary path to learn from/for physical data.

The method can be summarized as follows: Learn the model $F_\theta$ that fits the data constraints (regression, gan etc. ...), hence the solution at “grid points” $\mathbf{u}$. Then fit a simple function of separate space and time $u_f (t,x) = \alpha_j(t) \phi_j(t)$. (linear, quadratic, etc..) to interpolate between the prediction on which can enforce specific constraint, then a fortiori constraining the learned $u$. The experiments seem to support the author’s choices. The experiments are well conducted and present various use cases useful for the whole community.

However, the main weakness of the paper is the clarity in the presented interpolation method. Mainly, how it is computed and its link to the learned solution $\mathbf{u}(t)$. Also the estimation of $\phi$ is only very lightly discussed.

Question:

1. How well can the interpolant $u_f$ approximate the original to the problem ? Indeed, I guess this is related to the convergence of a numerical scheme: if the space is well covered (i.e. discrete step size goes down to 0) can we expect to recover a good function $u_f$ using such a prior?
2. Why piecewise cubic approximations do not require extra conditions (do you use Hermite polynomials and continuity of the derivative) compared to piecewise quadratic ? I suggest the authors include a thorough discussion on this topic in the appendices (with the conditions imposed on such local polynomials).
3. For the gan experiments, does the generator depend on time ? If  not, then the interpolation simply concerns the $\phi$ part ?
4. How to enforce a second order differential constraint when the interpolation is PWL ?

Remark on the presentation:
Despite interesting ideas, there is room for improving the presentation and the clarity of the paper.

a) Since the main focus of the experiments concerns “soft constraints”, I suggest the authors develop the section treating the soft-constraint components (Perhaps including an algorithm to show practical use) and how constraining $u_f$ constrains the learned $\mathbf{u}$ (or more specifically how the losses on $u_f$ impacts the $\theta$ of $F_\theta$), by detailing explicitly the link between the interpolation and the learned solution.


b) The key point of the paper lies in the interpolation scheme, however, in my opinion the paper lacks elements of details for an in depth understanding of the estimation method of the parameters $\alpha_j$ and $\phi$. Since this point is central in the paper, I highly recommend including thorough details on the estimation of the $\alpha$ and $\Phi$ at least in the appendices.

Minor:
 - I suggest for clarity to write the learned solution $u_\theta$ instead of $u$ to avoid the possible confusion between $u$ the solution to (1) and $\mathbf{u}$ the learned numerical solution at the nodes $x_1, .., x_N$).
- Should $t$ be a variable of $F_\theta$ in eq.2. Does $u_1 = u(x_1)$. ?
- Eq.6. If there are $N$ points ($x_1$, …, $x_n$). Should the sum stop at $N-1$ ?
- P.6 “with a multiplayer perceptron”


**Summary Of The Paper:**

The paper proposes a two-folded method to enforce constraints of different natures (differential, integrals….) on a statistical model learned from physical data.
The constraints are not enforced directly on the model but rather on “interpolant” functions that aims at completing the original model in between the observed grid points.


**Summary Of The Review:**

The paper proposes a very interesting method to enforce differential / Integral and differential constraints on an interpolation function derived from the learn operator that fits the data.
The experiments are well conducted but the method’s presentation lacks clarity (see main review) notably on the estimation of the interpolation coefficients and their link to the learned solution function?

If my concerns are addressed, I will be glad to increase my score.

---

> ### Author Response · Authors · 2021-11-17
> **Response to Reviewer 3bQB.**
>
> Thank you for your review.
>
> While the reviewer correctly describes the main ideas and incentives of our work, we believe there is some confusion regarding the method. Below, we provide a more detailed description of the method. We hope it will make it easier for the reader to grasp main details and, if this description proves to be clearer, we will be glad to add it to the main text.
>
> We consider a spatio-temporal system with state u(t, x) which satisfies a constraint h(u(t, x)) = 0, where t is time and x is position. We train a model to predict u(t, x) at points (x1, ..., xN) and, during training, force it to satisfy the constraint. Note that the model is trained to predict the data and satisfy the constraint simultaneously, i.e. all of the training occurs in one stage.
>
> The main problem here is how to evaluate h(u(t, x)). Evaluating constraints on uniform spatial grids is not a problem (e.g. using finite differences). Things get more tricky on unstructured grids though and none of the currently available methods can be applied here.
>
> Our solution is to approximate u(t, x) by u_f(t, x) using Eq. 6 (see Fig. 1) and then approximate h(u(t, x)) by h(u_f(t, x)). The basis functions \phi_i(x) are selected in advance and remain fixed while the coefficients \alpha_i(t) are obtained from the model's predictions as shown in Section 3.4 (last two paragraphs). Since we now have access to a function u_f(t, x), we can easily evaluate various constraints directly on u_f(t, x) instead of relying on observations at (x1, ..., xN) alone (as done with finite differences, for example). Note that we interpolate only over x and construct an interpolant for each time point t_i separately (i.e. we do not interpolate the states over time).
>
> We provide details of a single training iteration with soft constraints below:
> We denote model's predictions by u_pd and data by u_gt with the initial condition u0=u_gt[0]. Then, one training iteration is:
>
> 1) u_pd = predict(u0)  # Make prediction (Eq. 3)
> 2) u_f = interpolate(u_pd)  # Construct interpolant (Eq. 6)
> 3) constraint_violation = r(h(u_f))  # Evaluate constraint violation (Section 3.2)
> 4) loss = mse(u_pd, u_gt) + constraint_violation
> 5) Evaluate gradient of the loss and update model's parameters
>
> Below, we address the raised questions and concerns.
>
> **Q1:** How accurate is the interpolation?
> **A1:** Accuracy depends on two factors: node density and basis functions. Decreasing distances between nodes leads to reduced truncation error and more accurate approximation. Changing basis functions can also lead to reduced approximation error (e.g. replacing PWL basis with PWQ).
>
> **Q2:** Why piecewise cubic approximations do not require extra conditions as compared to PWQ basis?
> **A2:** We do not use piecewise cubic basis but rather the cubic rbf basis i.e. we set \phi_i(r) = r^3, so this basis does not require any extra conditions. We clarified this point in the manuscript (see blue updates).
>
> **Q3:** For the gan experiments, does the generator depend on time ?
> **A3:** No, in all experiments with GANs we generate static fields. Application of our method to time-dependent GANs is straightforward - simply apply constraints to each generated field at different time points.
>
> **Q4:** How to enforce a second order differential constraint when the interpolation is PWL ?
> **A4:** It cannot be done since PWL functions have zero second derivatives. If second, or higher, order derivatives are required, PWQ and other appropriate bases should be used (which is fully supported by our method).
>
> **Q5:** I suggest for clarity to write the learned solution $u_\theta$ instead of $u$ to avoid the possible confusion between $u$ the solution to (1) and $\mathbf{u}$ the learned numerical solution at the nodes $x_1,...,x_N$.
> **A5:** We do not learn the solution directly but rather the dynamics function $F_\theta$ which makes a prediction $\mathbf{u}(t)$ of the data $\mathbf{y}(t)$ starting from $\mathbf{y}(0)$ (Section 2).
>
> **Q5:** Should t be a variable of $F_\theta$ in eq.2? Does $u_1 = u(x_1)$?
> **A5:** While the state $u(t, x)$ does depend on time, the model we use assumes that the partial derivative of $u(t, x)$ wrt $t$ is stationary hence $F_\theta$ does not include time. However, extension to non-staionary model is straightforward. And yes, $u_1$ denotes the state at the position $x_1$ (we clarified it in the text, see blue changes).
>
> **Q6:** Eq.6. If there are $N$ points $x_1, ..., x_N$, should the sum stop at $N-1$?
> **A6:** No. We place a basis function at each observation point and the sum runs over all the basis functions (Section 3.1).

---

> > ### Comment · Reviewer_3bQB · 2021-11-22
> > **Regarding the Authors response**
> >
> > I acknowledge the authors' response and thank them for the clarification regarding the method.
> >
> > In the light of these explanations, I believe that the authors contribution is interesting, yet I concur with Reviewers (Aj9W, vTXr) regarding the lack of experimental comparison harming the applicability of the method.
> > Indeed, even a simple approximation of the space derivative with finite differences would have been an interesting benchmark.
> >
> > Therefore, i believe that, in its present form, the paper is borderline and maintain my original score.

---

> > > ### Author Response · Authors · 2021-11-22
> > > **Response to Reviewer 3bQB**
> > >
> > > Thank you for your comments.
> > >
> > > Please note that we provide comparisons of finite differences and our method (in two different settings) in Section 4.1 and show that both approaches produce very similar results.

---

### Official Review · Reviewer_A9JW · 2021-10-31

**Correctness:** 3
**Technical Novelty And Significance:** 2
**Empirical Novelty And Significance:** 2
**Recommendation:** 5
**Confidence:** 4

**Main Review:**

In general, the paper is easy to read. Many details remain vague though, e.g., the implementation of the constraints in the experiments, how to train the differentiable hard constraint solver, how exactly are constraints split into gamma_n terms (and why, this seems unnecessary), what is a thin plate spline and why is this a good basis for evaluating constraints.
Section 3.1, 3.2 could have been framed better by noting that this is simply a finite element formulation of constraints, and differential/integral formulations directly correspond to FE differentials/integrals.

As the paper mentions in the introduction, using soft constraints (often on regular grids) in training is an extremely common practice in physics prediction papers. Often, this is not referred to as constraints but as "physics-based"/auxiliary losses (e.g. adding a loss on divergence when modeling incompressible flow), but it amount to the exact same thing.
This paper differentiates itself from these methods by a) using unstructured grids, and b) providing a formalism for solving hard constraints.

However, the paper doesn't make a particular strong case for these choices. First, the results are either on regular grids (4.1), or irregular meshing of a square domain, but with a very uniform edge lengths (4.2-4.3). None of these demonstrate any advantage of irregular meshing, and I expect all examples could have been much more easily solved on regular grids with standard CNN-based methods.
And second, hard constraints are presented as one of the contributions, and the description spans quite a bit of the method section. However as the authors state themselves, their method for solving hard constraints is only feasible for very small systems, and is hence not used for any of the paper's main results (only for a minor example in the appendix).
Similarly, the paper introduces mitigation techniques to make quadratic basis functions (PWQ) work, which turn out to perform strictly worse than a simple linear basis in all settings.

And finally, I'm a bit unsure what to take away from the results. It's a bit of an obvious finding that putting an auxiliary loss on a certain quantity (e.g. divergence) will produce results with lower values for this quantity. It would have been much more interesting to show examples that actually highlight the contributions of the paper-- i.e. that isn't easily achieved with corresponding loss terms on regular grids. Also, this paper really needs comparisons to other methods, e.g. PDEs on regular grids with physics-based losses, or perhaps comparing constraints on energy to Hamiltonian methods, etc.

Other comments:
- On regular grid in 4.1, wouldn't the used PWL basis be exactly equivalent to FD?
- While there seems to be a bit less noise, I'm not sure that the constrained look 'more similar' to the data in fig 6, 9
- Why does the GNN in 4.2 not work? GNNs generally work very well for such PDEs. An MLP is likely a poor choice for learning local PDE dynamics, particularly when trained on small datasets

**Summary Of The Paper:**

The paper introduces and studies several variants of introducing soft and hard constraints into learned models of PDEs on unstructured meshes.

**Summary Of The Review:**

While I like aspects of this paper (e.g. I think the FE formulation for constraints on unstructured grids can be useful), in its current form it's not ready for publication. Many of the more interesting contributions (e.g. higher-order basis functions, hard constraints, irregular grids) don't really work that well and aren't used in the main experiments: most results are on square, regular domains, with soft constraints and linear or radial basis functions, which is quite similar to what a lot of physics prediction papers are already doing with auxiliary losses (and for which there aren't any baseline comparisons).
I'd encourage the authors to work further on the method (e.g. make hard constraints work for nontrivial systems), and choose good experiments to showcase the strengths of constraints on irregular grids. I also think constraints for adversarial methods that the paper touches on could be worth exploring further.

---

> ### Author Response · Authors · 2021-11-17
> **Response to Reviewer A9JW. Part 1.**
>
> Thank you for your review. Below, we address the raised questions and concerns.
>
> **Q1:** How were constraints in the experiments implemented?
> **A1:** Every experiment contains a description of the constraint it uses, while Section 3 describes how any constraint can be implemented and evaluated. In practice, evaluation of a constraint comes down to evaluating a matrix-vector product where the matrix is a constraint matrix \Phi (as shown Section 3) and the vector is the coefficients \alpha(t) obtained from the system's state u(t, x) (as shown in Section 3.4).
>
> **Q2:** How to train the differentiable convex program solver used to implement hard constraint?
> **A2:** There is no need to train it. We only need to be able to differentiate the output of the optimization procedure. To do that, we use the method from [1].
>
> **Q3:** How exactly are constraints split into gamma_n terms and why?
> **A3:** Please see examples in Section 3.1 Pointwise/Differential constraints. Splitting into gamma_n allows to formulate a general constraint evaluation technique that significantly simplifies the implementation.
>
> **Q4:** What is a thin plate spline and why is this a good basis for evaluating constraints?
> **A4:** Thin plate spline is an interpolation technique obtained by using \phi_i(r) = r^2 \log r in Equation 6 as the basis. We performed testing using thin plate spline due to its strong interpolating capabilities. The reason why testing was performed using a basis that was not used during training is to avoid biasing or ovefitting. We added these details to the manuscript (see green updates).
>
> **Q5:** Section 3.1, 3.2 could have been framed better by noting that this is simply a finite element formulation of constraints, and differential/integral formulations directly correspond to FE differentials/integrals.
> **A5:** While we agree that there is a minor resemblance with the finite element method (FEM) (namely, the fact that FEM also uses finite-dimensional function spaces to approximate a function), we do not necessarily agree that framing our method from that point of view would be beneficial. There are a few reasons for that. FEM is much less well known than interpolation and would require its own rather lengthy exposition. Also, FEM is tangentially related to interpolation and is rather focused on approximately solving weak forms of PDEs. Considering that the main focus of our method is interpolation, we believe presenting it in terms of FEM would seem a bit far fetched.
>
> **Q6:** This paper differentiates itself from previous methods by a) using unstructured grids, and b) providing a formalism for solving hard constraints.
> **A6:** We agree, but in addition to a) and b), it is also important to add that none of the previous methods allow to implement such a wide range of constraints as our method.
>
> **Q7:** The results are either on regular grids (4.1), or irregular meshing of a square domain, but with a very uniform edge lengths (4.2-4.3). None of these demonstrate any advantage of irregular meshing, I expect all examples could have been much more easily solved on regular grids with standard CNN-based methods.
> **A7:** Our goal was not to show advantages of using unstructured grids (of which there are many), but rather propose a method for enforcing constraints on such grids. Considering the importance and ubiquity of unstructured grids, we believe that having a method for enforcing a wide range of constraints on such grids would be useful. The fact that we use square domains plays no significant role as the grids are still unstructured. And indeed, CNN-based methods could be used here to learn the dynamics, but only if the data was available on a uniform grid.
>
> **Q8:** Hard constraints work only for small systems.
> **A8:** While our approach to enforcing hard constraints is currently limited to small-scale problems, this limitation is not fundamental and is mostly dependent on efficient implementations of differentiable convex optimization methods. The ability to satisfy constraints exactly might be important for some cases, thus making our hard constraint enforcing method potentially very valuable.
>
> **Q9:** The paper introduces mitigation techniques to make quadratic basis functions (PWQ) work, which turn out to perform strictly worse than a simple linear basis in all settings.
> **A9:** Indeed, the PWQ basis performs worse than the linear basis in this case. However, as Fig. 4 shows, it still can be on par or better than the unconstrained model which is already promising.

---

> > ### Author Response · Authors · 2021-11-17
> > **Response to Reviewer A9JW. Part 2.**
> >
> > **Q10:** This paper really needs comparisons to other methods.
> > **A10:** We would like to re-emphasize that, to the best of our knowledge, previous methods implement only selected constraint types and use finite differences which limits them to uniform grids, whereas we use interpolants which allows us to evaluate constraints on unstructured grids. As mentioned already above, in Section 4.1 we show that currently existing methods and our new method are accurate and result in similar performance in the setting where current methods can be applied. In Sections 4.2-4.4, we demonstrate the performance of our method in interesting cases that involve unstructured grids and a variety of different constraints. These represent cases where the currently available methods cannot be applied. The results show that our approach is simply more general and flexible, i.e., it allows to implement a wider range of constraints on a wider range of grid types.
> >
> > **Q11:** On regular grid in 4.1, wouldn't the used PWL basis be exactly equivalent to FD?
> > **A11:** Not quite. Finite differences allow to approximate derivatives only at nodes, while PWL functions are differentiable almost everywhere (except the nodes and cell boundaries).
> >
> > **Q12:** While there seems to be a bit less noise, I'm not sure that predictions of constrained models look 'more similar' to the data in fig 6, 9.
> > **A12:** We respectfully disagree with this statement. Figure 6 clearly shows that predictions of the constrained model are noticeably more monotone in the horizontal direction due to much smoother contour lines thus making the predictions more similar to the data. Similarly, Figure 9 shows even more dramatic difference between constrained and unconstrained models. The visual observations are supported by quantitative results shown in Fig. 7, 8.
> >
> > **Q13:** Why does the GNN in 4.2 not work? GNNs generally work very well for such PDEs. An MLP is likely a poor choice for learning local PDE dynamics, particularly when trained on small datasets.
> > **A13:** Indeed, this is an interesting question but we do not have a definitive answer as of yet. We tried various architectural options but it did not result in any improvements. GNNs, as many recent works have shown, can learn complex dynamics. Also, it can be noted that most works using GNNs to learn complex systems train them on short trajectories. Therefore, we can speculate that failure of our GNN to learn the dynamics of the Cahn-Hilliard equation can be attributed to how GNNs interact with continuous time dynamics, long training trajectories and adaptive solvers. MLPs seem to be easier to optimize in this kind of setting, but this is clearly a sub-optimal choice for large grids and small data regimes.
> >
> > **Q14:** I'm a bit unsure what to take away from the results.
> > **A14:** Main goals of the experimental section were: 1) to demonstrate that our method performs on par with currently existing method in the setting where current methods are applicable, and 2) demonstrate interesting use cases where our method can be applied and current methods cannot.
> >
> > Below, we summarize how these goals were achieved and highlight main points of the experimental results.
> >
> > First, in Section 4.1, we show that our method performs on par with currently existing methods, while at the same time being much more general and flexible (namely, our method allows to implement a wider range of constraints on a wider range of grid types).
> >
> > Then, in Sections 4.2-4.4, we demonstrate how learning of dynamical systems and GANs on unstructured grids benefits from physics-based constraints. Crucially, while current methods are only applicable to uniform grids, our method allows analyzing data that is collected on any unstructured grid and also expands the range of available types of constraints.
> >
> > In other words, constraints are important and should be enforced. Our methods allows to do that for a much wider range of constraints and grid types than any currently available method. We consider our contribution to be a significant improvement over currently available constraint evaluation methods.
> >
> > References
> > [1] [https://arxiv.org/abs/1910.12430](https://arxiv.org/abs/1910.12430)

---

> > > ### Comment · Reviewer_A9JW · 2021-11-23
> > > **Updated review**
> > >
> > > Thank you for the response, which clears up some of my questions regarding the paper's exposition.
> > >
> > > I still think the paper needs to make a stronger case demonstrating that the introduced constraints that go beyond what is commonly used are actually useful. I also disagree with the statement that "these represent cases where the currently available methods cannot be applied"; this is only true for the chosen unstructured discretization, but for the problems studied there is really no reason to do so, a regular grid discretization would work just as well or better, and (soft) FD constraints are trivial to formulate in this case.
> > >
> > > That said, while I won't change my score, I do think there is some value here. As I stated in my initial review I would encourage you to keep working on this; if you can show experiments where the nonstandard constraint formulations matter (e.g. problem domains that can't easily discretized on regular grids, problems where higher-order constraints show significant advantage etc.) and maybe finding a way to apply hard constraints to larger problems, this would be a much stronger paper.

---

### Official Review · Reviewer_vTXr · 2021-11-02

**Correctness:** 3
**Technical Novelty And Significance:** 2
**Empirical Novelty And Significance:** 2
**Recommendation:** 5
**Confidence:** 4

**Details Of Ethics Concerns:**

No ethics statement provided

**Main Review:**

Advantages

- The submission is written well, is clear up until the experimental section.

- The experiments show that the addition of constraints help to learn a model that does not violate said constraints.

- The central method is sound and simple enough to be reimplemented. That said, I think some experimental details are missing and I could not reimplement the experiments. That said, since code will be released, this is not such a big issue.

Queries

- Equation 1: can dynamics F also explicitly depend on t? As written, it does not.

- Equation 7: I’m confused about the dimensionality of \alpha_j. In Equation 6, it is a p-dimensional quantity, because u(t, x) \in R^p. Is it the case that u(t, x) is now scalar-valued, since h: R \to R and \gamma_m: R \to R? This should be made explicit in the text.

- Equation 10: what is g?

- Section 4.1: I think it would be more helpful to the reader if the experimental setups from [37] and [10] were described more completely in the main text. It is difficult to gauge from what is written, what exactly is going on and whether your results are meaningful

- Section 4.2: Could you provide intuition for why the MLP would work better than the GNN? I would hazard a guess that the GNN does not have a large enough receptive field. Since you only use a single layer of message passing from what I see in the appendix, the receptive field of an output neuron is the 1-step neighborhood due the the Delaunay triangulation. By contrast, the MLP receives the full domain as its receptive field.

- There seems to be a lack of baselines in the experimental section after section 4.1. I am not entirely sure why this is. Given that it is mentioned in the related work that there is a rich literature on incorporating constraints, both hard and soft, into learnable PDE models, I would expect to have seen some in the experiments.

- What do the error bars represent in Figures 4 and 7?

Minor notes

- Equation 2: the ordering of x and t in du(x, t)/dt does not align with elsewhere in the submission, where you have used u(t, x).

- Equation 2: Please be more specific what the notation F_\theta(x_i, x_N(i), …) means. For instance what is x_N(i) or u_N(i)? My understanding is that you have replaced the differential operator F with a local parametrised operator F_\theta over neighbourhoods N(i). The notation and underlying assumptions here should be explained more precisely.

**Summary Of The Paper:**

The authors present a method to incorporate constraints into the output of learnable PDE models. They cover point-wise, differential, and integral constraints. They achieve this by representing the PDE solution in a basis, as is common for the variational method (e.g. pseudo-spectral method, and finite element method). The neural network outputs interpolation coefficients for each time step into the future. To implement the constraints, they have two methods: ‘soft constraints’, whereby a constraint-breaking penalty is applied to the neural network output and ‘hard constraints’, whereby the output of the network is projected on to constraint-satisfying solutions. This latter method is achieved by solving a convex programme.

The authors test on a variety of benchmarks, demonstrating that their method works; although, it is hard to parse the experimental section for whether these results are significant.

**Summary Of The Review:**

I think the method is sound, and up until the experimental section the paper is well laid out and straight-forward to read. My understanding began to breakdown in the experimental section, where information was either referenced in the Appendix or completely in other papers. This made it hard to read and slowly down reviewing a lot. In most of the experiments, comparisons to other methods in the literature are missing and it is therefore hard to gauge the significance of this work.

---

> ### Author Response · Authors · 2021-11-17
> **Response to Reviewer vTXr. Part 1.**
>
> Thank you for your review. Below, we address the raised questions and concerns.
>
> **C:** It is hard to understand whether the experimental results are significant.
> **A**:  Main goals of the experimental section were:
>
> 1. To demonstrate that our method performs on par with currently existing method in the setting where current methods are applicable
>
> 2. To demonstrate interesting use cases where our method can be applied and current methods cannot
>
> Below, we summarize how these goals were achieved and highlight main points of the experimental results.
>
> First, in Section 4.1, we show that our method performs on par with currently existing methods, while at the same time being much more general and flexible (namely, our method allows to implement a wider range of constraints on a wider range of grid types).
>
> Then, in Sections 4.2-4.4, we demonstrate how learning of dynamical systems and GANs on unstructured grids benefits from physics-based constraints. Crucially, while current methods are only applicable to uniform grids, our method allows analyzing data that is collected on any unstructured grid and also expands the range of available types of constraints.
>
> In other words, constraints are important and should be enforced. Our methods allows to do that for a much wider range of constraints and grid types than any currently available method. We consider our contribution to be a significant improvement over currently available constraint evaluation methods.
>
> *Other points:*
>
> Our formulation naturally allows to enforce hard constraints via projections (Section 3.3). While this approach is currently limited to small-scale problems, this limitation is not fundamental and is mostly dependent on efficient implementations of differentiable convex optimization methods. The ability to satisfy constraints exactly might be important for some cases, thus making our hard constraint enforcing method particularly valuable.
>
> We demonstrate the potential of using higher-order Lagrange basis functions (PWQ) which are required if higher-order derivatives are present in the constraints. This is important because Lagrange bases are sparse and allow for efficient construction of the interpolant which makes them scalable to grids with large number of nodes.
>
> ### Questions
>
> **Q1:** Equation 1: can dynamics F also explicitly depend on t? As written, it does not.
> **A1:** Yes. Our method can be applied to arbitrary models. Although the results shown in the manuscript are obtained with a stationary dynamics model, extension to non-stationary models is straightforward.
>
> **Q2:** Equation 7: I’m confused about the dimensionality of $\alpha_j$. In Equation 6, it is a p-dimensional quantity, because $u(t, x) \in R^p$. Is it the case that $u(t, x)$ is now scalar-valued, since $h: R \to R$ and $\gamma_m: R \to R$? This should be made explicit in the text.
> **A2:** Yes, you are correct. In order to avoid cluttered notation, we used a scalar-valued $u(t,x)$ which was stated in the paragraph right above "Pointwise constraints" but we'll revise the text to make it more explicit.
>
> **Q3:** Equation 10: what is g?
> **A3:** As said at the beginning of the section, g is a constraint function.
>
> **Q4:** Section 4.1: I think it would be more helpful to the reader if the experimental setups from [37] and [10] were described more completely in the main text. It is difficult to gauge from what is written, what exactly is going on and whether your results are meaningful
> **A4:** We agree. Although we provide a more complete description in the appendices, we will revise this section and add as much information in the main text as the page limit permits.
>
> **Q5:** Section 4.2: Could you provide intuition for why the MLP would work better than the GNN? I would hazard a guess that the GNN does not have a large enough receptive field. Since you only use a single layer of message passing from what I see in the appendix, the receptive field of an output neuron is the 1-step neighborhood due the the Delaunay triangulation. By contrast, the MLP receives the full domain as its receptive field.
> **A5:** Indeed, this is an interesting question but we do not have a definitive answer as of yet. We tried increasing the receptive field of the GNN to two neighbors (which should be enough since the CH equation contains only fourth order derivatives), but it did not improve the results. Addition of more GNN layers leads to extremely slow forward and backward passes and ultimately to no improvements. GNNs, as many recent works have shown, can learn complex dynamics. Also, it can be noted that most works using GNNs to learn complex systems train them on short trajectories and in a discrete time setting. Therefore, we can speculate that failure of our GNN to learn the dynamics of the Cahn-Hilliard equation can be attributed to how GNNs interact with continuous time dynamics, long training trajectories, adaptive solvers and complex dynamical systems.

---

> > ### Author Response · Authors · 2021-11-17
> > **Response to Reviewer vTXr. Part 2.**
> >
> > **Q6:** There seems to be a lack of baselines in the experimental section after section 4.1. I am not entirely sure why this is. Given that it is mentioned in the related work that there is a rich literature on incorporating constraints, both hard and soft, into learnable PDE models, I would expect to have seen some in the experiments.
> > **A6:** We would like to re-emphasize that, to the best of our knowledge, previous methods implement only selected constraint types and use finite differences which limits them to uniform grids, whereas we use interpolants which allows us to evaluate constraints on unstructured grids. As mentioned already above, in Section 4.1 we show that currently existing methods and our new method are accurate and result in similar performance in the setting where current methods can be applied. In Sections 4.2-4.4, we demonstrate the performance of our method in interesting cases that involve unstructured grids and a variety of different constraints. These represent cases where the currently available methods cannot be applied. The results show that our approach is simply more general and flexible, i.e., it allows to implement a wider range of constraints on a wider range of grid types.
> >
> > **Q7:** What do the error bars represent in Figures 4 and 7?
> > **A7:** One standard deviation of the results over multiple random seeds. We updated the manuscript to include this information (see red updates).
> >
> > **Q8:** Equation 2: the ordering of $x$ and t in $du(x, t)/dt$ does not align with elsewhere in the submission, where you have used $u(t, x)$.
> > **A8:** Indeed. We updated the manuscript.
> >
> > **Q9:** Equation 2: Please be more specific what the notation F_\theta(x_i, x_N(i), …) means. For instance what is x_N(i) or u_N(i)? My understanding is that you have replaced the differential operator F with a local parametrised operator F_\theta over neighborhoods N(i). The notation and underlying assumptions here should be explained more precisely.
> > **A9:** We agree. Briefly, x_N(i) and u_N(i) are defined as sets of neighbors' positions and states. We updated the manuscript to include this information (see red updates).

---

> > > ### Comment · Reviewer_vTXr · 2021-11-25
> > > **Response to author rebuttal**
> > >
> > > _Q1: Equation 1: can dynamics F also explicitly depend on t? As written, it does not._
> > >
> > > Thanks for addressing this
> > >
> > > _Q2: Equation 7: I’m confused about the dimensionality of..._
> > >
> > > Thanks for making this explicit in the text
> > >
> > > _Q3: Equation 10: what is g?_
> > >
> > > Would be useful to mention this where and when you make use of it
> > >
> > > _Q4: Section 4.1: I think it would be more helpful to the reader if the experimental setups from [37] and [10] were described more completely in the main text. It is difficult to gauge from what is written, what exactly is going on and whether your results are meaningful_
> > >
> > > Thanks for this change
> > >
> > > _Q5: Section 4.2: Could you provide intuition for why the MLP would work better than the GNN? I would hazard a guess that the GNN does not have a large enough receptive field. Since you only use a single layer of message passing from what I see in the appendix, the receptive field of an output neuron is the 1-step neighborhood due the the Delaunay triangulation. By contrast, the MLP receives the full domain as its receptive field._
> > >
> > > Thanks for this. I might add though that using 2 neighbours for 4th order derivatives seems awful small. Higher accuracy stencils are usually larger than this and pseudo-spectral methods use the whole domain, which is why they are considerably more robust as a derivative estimation technique.
> > >
> > > _Q6: There seems to be a lack of baselines in the experimental section after section 4.1. I am not entirely sure why this is. Given that it is mentioned in the related work that there is a rich literature on incorporating constraints, both hard and soft, into learnable PDE models, I would expect to have seen some in the experiments._
> > >
> > > I see that the kinds of constraints are interesting. When you say that these have not been considered elsewhere in the literature are you referring to the ML literature or the numerical method literature?
> > >
> > > _Q7: What do the error bars represent in Figures 4 and 7?_
> > >
> > > Thanks for the amendment
> > >
> > > _Q8: Equation 2: the ordering of..._
> > >
> > > Thanks for the amendment
> > >
> > > _Q9: Equation 2: Please be more specific what the notation F_\theta(x_i, x_N(i), …) means. For instance what is x_N(i) or u_N(i)? My understanding is that you have replaced the differential operator F with a local parametrised operator F_\theta over neighborhoods N(i). The notation and underlying assumptions here should be explained more precisely._
> > >
> > > Thanks for the amendment

---

> > > > ### Author Response · Authors · 2021-11-26
> > > > **Response to Reviewer vTXr**
> > > >
> > > > **Q:** I might add though that using 2 neighbours for 4th order derivatives seems awful small.
> > > > **A:** To clarify, we increased the receptive field to two neighbors on each side, so four neighbors were used in total. Using four neighbors is the bare minimum for estimating a fourth order derivative, but it still can provide a second-order accurate estimate.
> > > >
> > > > **Q:** I see that the kinds of constraints are interesting. When you say that these have not been considered elsewhere in the literature are you referring to the ML literature or the numerical method literature?
> > > > **A:** All our claims concern only the ML literature. The method we use (evaluation of derivatives/integrals on an interpolant) is fundamental in numerical methods, but it has not yet been used in ML since models working on unstructured grids were introduced quite recently.

---

> > > > > ### Comment · Reviewer_vTXr · 2021-11-29
> > > > > **Final response**
> > > > >
> > > > > My thanks to the authors and other reviewers for their engagement in this process.
> > > > >
> > > > > I believe the kernel of the idea presented is interesting, effective, and novel within the presented context, but I also stand by my original review, and that of the other reviewers, that the empirical demonstration of this method needs improvement. This is a matter of thoroughness and rigour and will only serve to improve the current submission. If the submission is not accepted this time around, I would like to encourage the authors to continue working on this thread and to present a set of experiments, which clearly demonstrate both how their method can improve upon existing methods, and proper ablations into why.

---

### Decision · Program_Chairs · 2022-01-20

**Decision:**

Reject

**Comment:**

The paper introduces a framework for enforcing constraints into deep NNs used for modeling spatio-temporal dynamics characterizing physical systems. The authors consider different types of constraints (pointwise, differential and integral). They start from a formulation approximating PDEs as set of ODEs (method of lines). Their main idea is to approximate the solution of the equations using an interpolant between observations and imposing the constraints on this approximation function. The interpolant is built using basis functions located at observation points. The formalism considers irregular spatial grids and both soft and hard constraints. The main claim is then the introduction of a general formalism for considering different types of constraints on irregular grids. Experiments illustrate the behavior of the proposed method on different types of evolution equations and constraints.

The reviewers agree that the proposed approach is interesting and that some of the ideas are original. However, they also consider that the paper is not convincing enough to demonstrate the interest and novelty of the approach, compared to alternative methods. The experimental section mainly considers (except for one application) regular grids and constraints that could be handled by other methods as well. The authors should present cases where their method provides a clear advantage, distinct from existing solutions. The authors provided a well-argued rebuttal, clarifying several points. However, all reviewers retained their original scores and encourage the authors to further develop the experimental analysis to present a stronger paper. In addition, the presentation could be improved, and some technical aspects better explained (e.g., description of interpolation methods, and some advice on which interpolant to choose for a given problem).